# Benchmarking the Robustness of CNN-based Spatial-Temporal Models

**Chenyu Yi**[1*]    **Siyuan Yang**[1,2*]    **Haoliang Li**[3]    **Alex C. Kot**[1]

[1]School of Electrical and Electronic Engineering, Nanyang Technological University, Singapore
[2]Interdisciplinary Graduate Programme, Nanyang Technological University, Singapore
[3]Department of Electrical Engineering, City University of Hong Kong, China
{yich0003,siyuan005}@e.ntu.edu.sg    haoliali@cityu.edu.hk    eackot@ntu.edu.sg

## Abstract

The state-of-the-art deep convolutional neural networks are vulnerable to common corruptions in nature (e.g., input data corruptions caused by weather changes, system errors). While rapid progress has been made in analyzing and improving the robustness of models in image understanding, the robustness in video understanding is largely ignored. In this paper, we establish a corruption robustness benchmark, Mini Kinetics-C and Mini SSV2-C, which considers temporal corruptions beyond spatial corruptions in images. We make the first attempt to conduct an exhaustive study on corruption robustness in terms of spatial and temporal domain, using established CNN-based spatial-temporal models. The study provides some guidance on robust model design, training and inference: 1) 3D modules make video classification models more robust instead of 2D modules, 2) longer input length and uniform sampling of input frames can benefit model corruption robustness, 3) model corruption robustness (especially robustness in temporal domain) enhances with computational cost, which may contradict with the current trend of improving the computational efficiency of models. Our codes are available on https://github.com/Newbeeyoung/Video-Corruption-Robustness.

## 1   Introduction

The rapid progress has been witnessed in the past decade in both image and video understanding, mainly backup by the advances of convolutional neural networks (CNNs) and large-scale datasets. However, most of the datasets only consider clean data for training and evaluation purposes, while the models deployed in the real world will encounter common corruptions on input data such as weather changes, motions of the camera, and system errors [16][33]. Many works have shown that the image understanding models which capture only spatial information are not robust towards these corruptions [9][10][16]. While some efforts have been made to improve the model robustness under such corruptions for visual content based on spatial domain, the temporal domain information is largely ignored. Compared with images, the videos contain abundant temporal information, which can play an important role in video understanding tasks. The sensitivity of human vision towards corruptions is correlated with the temporal structure in videos [39], and the CNN models can also benefit from temporal information besides spatial information on the generalization of video understanding [17][43]. Therefore, analyzing the robustness of CNN models from the perspective of both spatial and temporal domain is crucial for getting the CNN models out to the real world.

In this paper, we make the first attempt to evaluate the corruption robustness on video content in terms of both spatial and temporal domains by proposing to answer the following research questions.

---

*equal contribution

Preprint. Under review.

1. how robust are the video classification models when they use temporal information?

2. how robust are the models against corruptions which correlate with a set of continuous frames, termed temporal corruptions, beyond the corruptions which only depend on content in a single frame, termed spatial corruptions (i.e., packet loss in the video stream can propagate error in the consequent frames)?

3. what is the trade-off between model generalization, efficiency and robustness?

Particularly, we propose two datasets (Mini Kinetics-C, Mini SSV2-C) based on current large-scale video datasets to benchmark the corruption robustness of models in video classification. We choose these two datasets as the benchmark for two reasons. Firstly, Kinetics [2] and SSV2 [13] are the most popular large-scale video datasets with more than 200K videos of each. Secondly, Kinetics relies on spatial semantic information for video classification, while SSV2 contains more temporal information [3][35]. It enables us to evaluate the corruption robustness of models on different types of datasets. To estimate the unseen test data distribution under the natural circumstance, we construct these two datasets by applying 12 types of corruptions which commonly happening in video acquisition and processing on clean official validation datasets. Each corruption contains 5 levels of severity. Different from image-based benchmarks [16][30][20][42], video-based datasets has another time dimension, so the corruptions we propose are separated into spatial level and temporal level. To estimate the distribution of corruptions in nature, we reduce the proportion of noise and blur, increase the corruption type caused by environmental changes and system error.

Based on the proposed robust video classification benchmark, we conduct large-scale evaluations on the corruption robustness of the state-of-the-art CNNs. For a fair comparison, all models are trained on clean videos, while evaluated on corrupted videos. We also examine the effect of architecture properties, training and evaluation protocols on corruption robustness. Based on the evaluation, we have several findings: a) 3D module makes the video classification model more robust instead of 2D module. b) Longer input frame length and uniform sampling of input can improve the corruption robustness. It implies that the input with more temporal information can make models more robust. c) The corruption robustness increases with the computation cost of models, which contradicts the current trend of improving the efficiency of models.

Our contributions can be summarized as follow: 1) we propose a robust video classification benchmark for evaluating the corruption robustness of models. Apart from spatial level corruptions in image-based robustness benchmarks, we take temporal corruptions into considerations in our benchmarks due to the temporal structure of videos. 2) We empirically evaluate the corruption robustness of common architecture properties, training and evaluation protocols. Based on the analysis involving over 60 models and 60 corruptions, we provide a guidance for robust model design, training and inference for video classification. 3) We also benchmark the state-of-the-art video classification models and draw conclusions on improving robustness of models in the future.

## 2 Related Work

### 2.1 Corruption Robustness of Neural Network

The vulnerability of deep neural networks has been widely studied in the past few years. The adversarial examples which are imperceptible to human being can lead to misclassification of models [12][28]. It demonstrates the vulnerability of neural networks against on-purpose adversarial attacks. Different from carefully crafted adversarial examples, common corruptions are more general in the nature. For example, the weather like rain, fog can cause corruptions on the input video or images; shot noise is caused by the discrete nature of light. Some works [9][10][11][16] show that common corruptions can degrade the deep neural networks performance significantly.

Hendrycks et al. [16] firstly proposed benchmarks ImageNet-C and CIFAR10-C to measure the corruption robustness of models against these unseen corruptions in the test dataset. In the following, the corruption robustness benchmarks are proposed for object detection [30], semantic segmentation [20] and pose estimation [42]. These image-based corruption robustness use the same 15 types of corruptions for evaluation. It is questionable that these corruptions estimate the distribution of corruptions existing in the real world. For example, corruptions in Noise and Blur categories occupy half of the total corruptions. Some works show that noise augmentation training can achieve the state-of-the-art corruption robustness [32]. In this work, we propose new temporal corruptions for

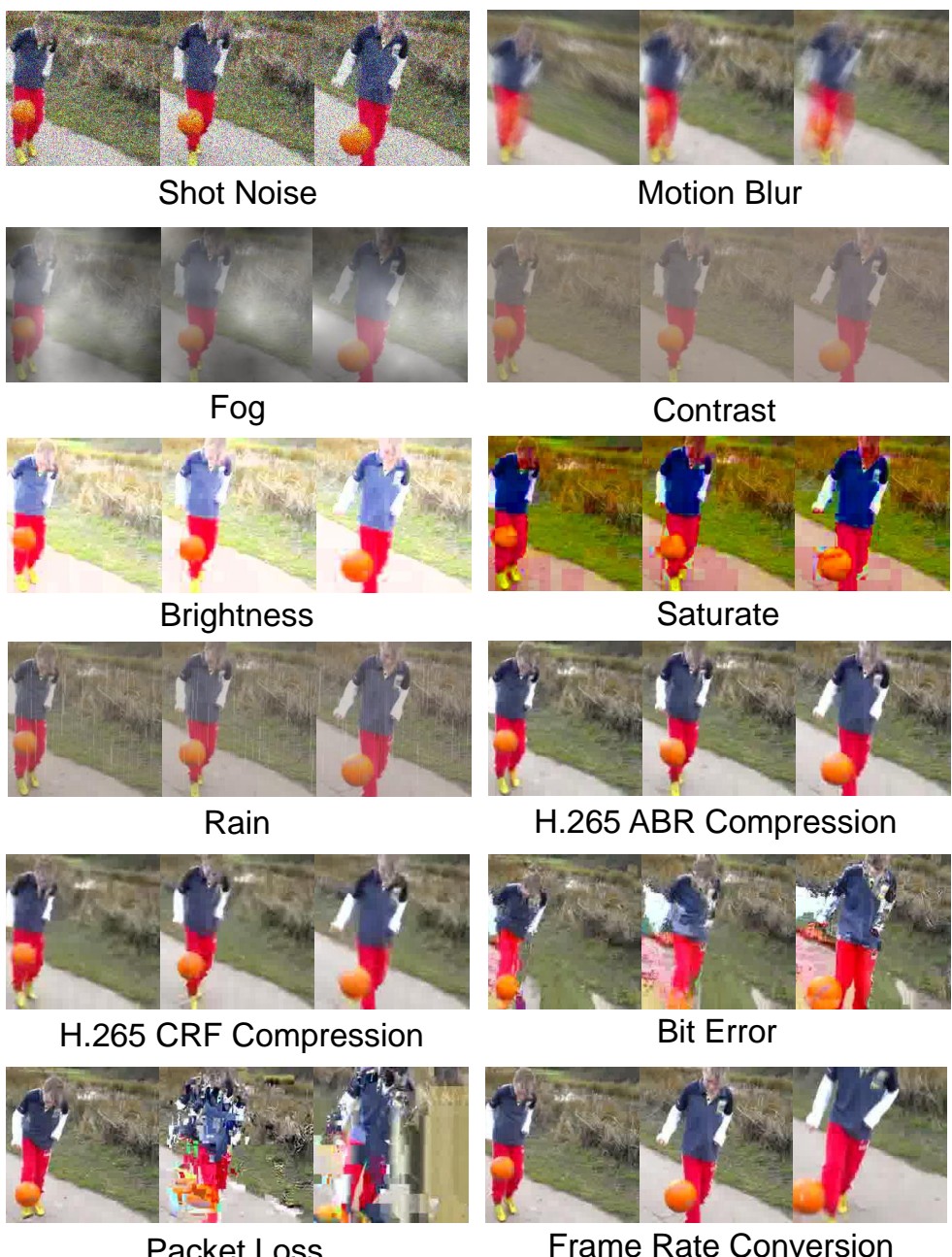

Figure 1: Our proposed corruption robustness benchmark consists of 12 types of corruptions with five levels of severity for each video. In the visualization examples, we use uniform sampling to extract 3 frames from each corrupted video, the sampling interval is 10 frames.

video-based classification and adjust the proportion of Noise and Blur corruptions. We are the first one to conduct exhaustive analysis on corruption robustness of deep convolutional neural networks in video classification.

## 2.2 Video Classification

Video classification has been one of the most active research areas in computer vision. Recently, with the great progress of deep learning techniques, various deep learning architectures have also been proposed. The advanced deep learning works for RGB-based video classification, which can

be mainly divided into two categories, namely, two-stream 2D CNN [4][8][7][25][36][43] and 3D CNN-based methods [2][5][6][19][31][37][40][41][45].

Two-stream 2D CNN framework first introduce by Simonyan and Zisserman [36], which proposed a two-stream CNN model consisting of a spatial network and a temporal network. Wang et al. [43] divided each video into three segments and processed each segment with a two-stream network. Fan et al. [4] presented an efficient and memory-friendly video architecture to capture temporal dependencies across frames. Plenty of works have extended 2D CNNs to 3D structures to simultaneously model the spatial and temporal context information in videos that is crucial for video action classification. Tran et al. [40] proposed a 3D CNN based on VGG models, named C3D, to learn spatio-temporal features from a frame sequence. Carreira and Zisserman [2] propose converting a pre-trained 2D ConvNet [38] to 3D ConvNet by inflating the filters and pooling kernels with an additional temporal dimension. Xie et al. [45] replaced many of the 3D convolutions with low-cost 2D convolutions to seek a balance between speed and accuracy. Feichtenhofer et al. [6] designed a two-stream 3D CNN framework containing a slow pathway and a fast pathway that operate on RGB frames at low and high frame rates to capture semantic and motion, respectively. More recently, Feichtenhofer et al. [5] also investigated whether the light or heavy model is required and presented X3D as a family of efficient video networks. In this paper, we establish the benchmark and extensively experiments to evaluate the robustness of these 2D CNN and 3D CNN methods.

## 3 Benchmark Creation

### 3.1 Corruption Robustness in Classification

When we deploy a computer vision system in the real world, the system may encounter different types of common corruptions caused by unforeseeable environmental changes (e.g., rain, fog) or system errors (e.g., network error). The performance of models under the corruptions is defined as corruption robustness [16]. It measures the average-case performance of models. To be more specific, the first line of corruption robustness considers the cases where the corruptions in the test data are unseen in the training. All the corruptions follow a distribution $P_C$ in the real world: $c \sim P_C$. For each type of corruption $c$, the corrupted samples $(c(x), y)$ follow the distribution $P_{(c(X),Y)}$: $(c(x), y) \sim P_{(c(X),Y)}$. For a classifier $f : X \to Y$, a general form of corruption robustness is then given by

$$R_f := \mathbb{E}_{c \sim P_C}[\mathbb{E}_{(c(x),y) \sim P_{(c(X),Y)}}[f(c(x)) = y)]], \tag{1}$$

where $x$ is the input, $y$ is the corresponding target label.

### 3.2 Metric Explanation

While it is impossible to evaluate the expectation in Equation 1 exactly by considering all corrupted data, we can estimate it using finite types of corruptions and limited corrupted samples. With the estimation, we use mean performance under corruption (mPC) to evaluate the corruption robustness of models in classification:

$$mPC = \frac{1}{N_c} \sum_{c=1}^{N_c} \frac{1}{N_s} \sum_{s=1}^{N_s} CA_{c,s}, \tag{2}$$

where $CA_{c,s}$ is the classification accuracy on dataset samples with corruption type $c$ under severity level $s$, $N_c$ and $N_s$ are the number of corruption types and the number of severity levels. The total number of corruptions applied on clean samples is $N_c \times N_s$. It follows the similar metric as [30]. Because the classification accuracy of models in video classification task highly relies on input length, frame rate, sampling methods and the architectures, the absolute classification accuracy cannot assess the robustness of models fairly under different settings. To evaluate the robustness of models based on their generalization ability on clean data, relative mean performance under corruption are also used in this paper which is given as

$$rPC = \frac{mPC}{CA_{clean}}, \tag{3}$$

where $CA_{clean}$ is the classification accuracy of models on video classification datasets without any corruption.

### 3.3 Benchmark Datasets

The robust video classification benchmark contains two benchmarks datasets: Mini Kinetics-C and Mini SSV2-C. In each dataset, we apply $N_c = 12$ types of corruptions with $N_s = 5$ level of severity on clean data to estimate the possible unseen test data. Most of the previous image-related benchmarks [16][30][42] use the same 15 types of corruptions to benchmark the general robustness without proposing new types of corruptions. Different from image-based tasks, video classification task uses video as the inputs. From the aspect of video processing and formation, corruptions can generate at various stages. We summarize the corruptions adopted for the proposed benchmark below.

**Video Acquisition:** *Shot Noise, Motion Blur, Fog, Contrast, Brightness, Saturate, Rain.*
**Video Processing:** *H.265 ABR Compression, H.265 CRF Compression, Bit Error, Packet Loss, Frame Rate Conversion.*

To be specific, shot noise [1][29], motion blur [18][22], fog [21][46], contrast, brightness, and saturate corruptions are presented in image-based corruption robustness benchmarks. Besides, we add rain corruption [26][27] in our benchmark because it is the most common weather in nature. The other five types of corruptions can only be generated in the pipeline of **video processing** [34][44]. We firstly introduce two types of corruptions caused by video compression. In many video-based application with communication network, the raw video requires high bandwidth but the bandwidth of network is limited. As a result, compression is compulsory for these real-world applications. The first type compression H.265 ABR compression use the popular codec H.265 for compression. The compression targets an Average Bit Rate, it is a lossy video compression which generates compression artifacts. Another type corruption is caused by H.265 CRF Compression. Different from Average Bit Rate, Constant Rate Factor (CRF) is another encoding mode by controlling quantization parameter. It introduces compression artifacts as well. Bit Error and Packet Loss corruptions come from video transmission due to imperfect channels in the real world [24]. The error will propagate in the consequent channel and cause more severe corruptions. Frame Rate Conversion means the frame rate of test data can differ from the frame rate of training data. Because of limited bandwidth, the video capturing system deployed in the wild will use lower frame rate for transmission. Specifically, we categorize motion blur, ABR compression, CRF compression, bit error, packet loss and frame rate conversion into temporal corruptions. These corruptions are correlated to a set of continuous frames. The rest of corruptions are categorized into spatial corruptions, because they only depend on the content in a single frame. We show the corruption examples in Figure 1.

**Mini Kinetics-C:** Kinetics [2] are the most popular benchmark for video classification task in the past few years. It contains 240K training videos and 20K validation videos of 400 classes. Each video in Kinetics lasts for 6-10 seconds. Considering the scale of Kinetics, it is more practical to use half of the dataset for training and evaluation. It enables us to conduct experiments on more settings with less infusion for video classification tasks. To create Mini Kinetics-C, We firstly randomly pick 200 classes from Kinetics to create Mini Kinetics. Then we apply 12 types of corruptions with 5 levels of severity on the validation dataset of Mini Kinetics, so Mini Kinetics-C is 12x5 times the validation dataset in Mini Kinetics.

**Mini SSV2-C:** SSV2 dataset [13] consists of 168K training videos and 24K validation videos of 176 classes. Each video lasts for 3-5s. Different from Kinetics, the SSV2 dataset highly relies on temporal information for classification. It also has less background information. Similar to Mini Kinetics-C, we randomly choose 87 classes from the original SSV2 and apply the corruptions on the validation dataset to create Mini SSV2-C.

## 4 Experiments

With the proposed corrupted video classification datasets, we raise several questions intuitively: **Q1:** how robust are current models? **Q2:** how robust are the models against spatial and temporal corruptions? **Q3:** Are the trends of increasing video classification model generalization ability and efficiency aligned with the target of improving their robustness?

We conduct a large-scale analysis involving more than 60 video classification models and the proposed two datasets. Each dataset contains 60 different corruptions. Based on the study, we answer the three questions in Section 4.2, 4.3 and 4.4 correspondingly. In Section 4.5, we extensively evaluate the impact of the architecture properties, the sampling methods, the input lengths and temporal

Table 1: Corruption robustness of architectures on the corrupted Mini Kinetics and Mini SSV2. All the models are trained with 32-frame inputs and evaluated at clip level. I3D and S3D use a backbone of InceptionV1. X3D-M, TAM and SlowFast use a backbone of ResNet50.

| Approach | Clean | mPC | rPC | Spatial | | | | | | Temporal | | | | | |
| --- | --- | --- | --- | --- | --- | --- | --- | --- | --- | --- | --- | --- | --- | --- | --- |
| | | | | Shot | Rain | Fog | Contrast | Brightness | Saturate | Motion | Frame Rate | ABR | CRF | Bit Error | Packet Loss |
| Mini Kinetics-C | | | | | | | | | | | | | | | |
| S3D [45] | 69.4 | 56.9 | **82.0** | 50.8 | 51.5 | 47.6 | 46.4 | 62 | 56.8 | 54.9 | 56.8 | 68.3 | 62.8 | 59.1 | 59.9 |
| I3D [2] | 70.5 | 57.7 | 81.8 | **56.1** | 50.5 | 45.8 | 46.3 | 63.1 | 57.5 | 56.5 | 68.9 | 64.3 | 60.8 | 59.4 | 62.9 |
| 3D ResNet-18 [14] | 66.2 | 53.3 | 80.5 | 47.7 | 45.8 | 40.5 | 43 | 58.9 | 53.5 | 53.4 | 65.1 | 60.1 | 56.4 | 55.8 | 59.3 |
| 3D ResNet-50 [14] | **73.0** | **59.2** | 81.1 | 49.6 | 47.6 | **49.1** | **51.5** | **65.8** | **60.0** | **59.3** | **71.9** | **65.9** | **61.6** | **62.2** | **65.9** |
| SlowFast 8x4 [6] | 69.2 | 54.3 | 78.5 | 33.4 | **51.7** | 40.7 | 46.5 | 61.2 | 54.1 | 54.9 | 68.5 | 63.1 | 59.1 | 56.7 | 62.2 |
| X3D-M [5] | 62.6 | 48.6 | 77.6 | 33.2 | 44.4 | 36.9 | 40.9 | 54.8 | 48.6 | 47.3 | 61.0 | 55.9 | 53.6 | 51.3 | 55.9 |
| TAM [4] | 66.9 | 50.8 | 75.9 | 47.6 | 47.8 | 35.8 | 38.2 | 58.0 | 45.3 | 44.5 | 65.0 | 58.1 | 54.3 | 55.9 | 59.7 |
| Mini SSV2-C | | | | | | | | | | | | | | | |
| S3D [45] | 58.2 | 47.0 | **81.8** | 40.3 | 27.5 | 43.3 | 48.1 | 52.3 | 47.7 | 36.1 | 47.5 | 53.2 | 52.4 | 43.8 | 50.8 |
| I3D [2] | 58.5 | **47.8** | 81.7 | **43.7** | 30.5 | **43.4** | 46.3 | 53.1 | **49.5** | **36.3** | 49.0 | 53.7 | 52.6 | 44.3 | 51.3 |
| 3D ResNet-18 [14] | 53.0 | 42.6 | 80.3 | 34.1 | 21.9 | 38.0 | 42.9 | 48.0 | 42.5 | 34.9 | 42.9 | 49.1 | 47.8 | 40.3 | 46.9 |
| 3D ResNet-50 [14] | 57.4 | 46.6 | 81.2 | 39.8 | 24.4 | 39.7 | 47.8 | 51.5 | 45.1 | 36.1 | 47.3 | 52.3 | 50.5 | 43.7 | 50.5 |
| SlowFast 8x4 [6] | 48.7 | 38.4 | 78.8 | 26.9 | 34.0 | 32.4 | 34.9 | 40.4 | 34.6 | 27.7 | 37.1 | 44.4 | 44.0 | 36.1 | 41.8 |
| X3D-M [5] | 49.9 | 40.7 | 81.6 | 32.9 | **37.0** | 36.6 | 39.4 | 44.2 | 38.8 | 28.5 | 39.55 | 46.0 | 45.9 | 38.4 | 43.9 |
| TAM [4] | **61.8** | 45.7 | 73.9 | 39.1 | 19.2 | 43.3 | **52.1** | **53.7** | 45.0 | 33.4 | **49.9** | **55.5** | **54.5** | **47.9** | **54.3** |

information on robustness towards common corruptions. To compare the models and settings fairly, we use the same training and evaluation protocol on all models.

## 4.1 Training and Evaluation

We follow the training protocol in [3]. For the models with 8-frame input, we initialize them with ImageNet-Pretrained weights. Then we fine-tune the models using longer input on the weights of models using shorter input. For example, we initialize I3D using 32 frames by the weights of I3D using 16 frames. Because the batch size has a large impact on the performance of models, we use a batch size of 32 to train most models. For some models using 64 frames, we slightly reduce the batch size to 24 due to GPU capacity limits. For data preprocessing and augmentation, we extract the image frame from videos and resize the shorter size to 240. We then apply multi-scale crop augmentation on each clip input and resize them to 224x224. We train all the models on clean datasets, using an initial learning rate of 0.01.

In video classification, the model can be evaluated at clip level and video level settings using uniform sampling and dense sampling. At clip level, we randomly choose one frame from the first segment of video, and extract frames with fixed stride for uniform sampling. For dense sampling, we randomly choose one frame from the whole video and use the following frames as input. At the video level, we extract first $N$ frames and use each frame as the first frame of each clip for uniform sampling, where $N$ is the number of clips used in video level uniform sampling. For dense sampling, we crop the video evenly and extract $M$ clips with continuous frames. In our experiments, we use $N = 4$ and $M = 8$ because the performance of uniform sampling saturates faster. For all types of corruption, we use the same settings to evaluate the models.

## 4.2 Q1: Benchmarking Robustness of SOTA Approaches

We trained several networks, including S3D, I3D, 3D ResNet, SlowFast, X3D and TAM on clean Mini Kinetics and Mini SSV2. S3D, I3D and 3D ResNet are the prior SOTA methods for video classification, while SlowFast, X3D and TAM claim to achieve the SOTA performance with less computational cost. We use ResNet50 as the backbone of SlowFast, X3D and TAM for a fair comparison. During training, we use uniform sampling to extract 32 frames as input. We then evaluated the models using uniform sampling at 32 frames at the clip level. The corruption robustness of the state-of-the-art spatial temporal models are shown in Table 1. It shows that 3D ResNet50 achieves the best performance on clean accuracy and mPC on Mini Kinetics-C. Although TAM performs the best on clean Mini SSV2, its mPC is 2.1% lower than I3D. S3D reaches the highest rPC on both datasets. When we increase the backbone size from ResNet18 to ResNet50 for 3D ResNet approach, the improvements of clean accuracy, mPC and rPC on both datasets indicate that the robustness increases with model capacity.

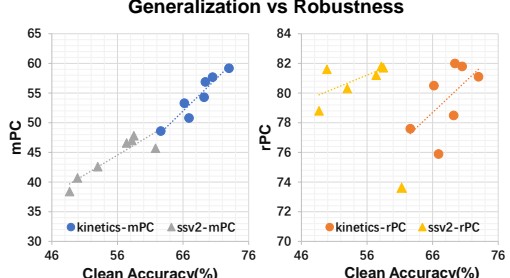

Figure 2: The generalization and robustness of the SOTA approaches shown in Table 1.

Figure 3: The efficiency and robustness of the SOTA approaches shown in Table 1.

Table 2: mPC and rPC on spatial and temporal corruptions

| Approach | Mini Kinetics-C | | | | Mini SSV2-C | | | |
| | spatial | | temporal | | spatial | | temporal | |
| | mPC | rPC | mPC | rPC | mPC | rPC | mPC | rPC |
|---|---|---|---|---|---|---|---|---|
| S3D [45] | 52.5 | **75.7** | 61.3 | **88.4** | 43.2 | 74.2 | 47.3 | 81.2 |
| I3D [2] | 53.2 | 75.5 | 62.1 | 88.1 | **44.4** | 75.9 | 47.9 | 81.8 |
| 3D ResNet-18 [14] | 48.2 | 72.9 | 58.3 | 88.1 | 37.9 | 71.5 | 43.7 | **82.4** |
| 3D ResNet-50 [14] | **53.9** | 73.9 | **64.5** | 88.3 | 41.4 | 72.1 | 46.7 | 81.4 |
| SlowFast 8x4 [6] | 47.9 | 69.3 | 60.7 | 87.8 | 33.9 | 69.5 | 38.5 | 79.1 |
| X3D-M [5] | 43.1 | 68.9 | 54.1 | 86.5 | 38.2 | **76.5** | 40.4 | 80.9 |
| TAM [4] | 45.5 | 67.9 | 56.2 | 84.0 | 42.1 | 68.1 | **49.3** | 79.7 |

### 4.3 Q2: Robustness w.r.t Spatial and Temporal Corruptions

**Robustness w.r.t spatial corruptions.** As we can observe from Table 1, models trained on Mini Kinetics are vulnerable to fog, contrast and shot noise, while models trained on Mini SSV2 are vulnerable to rain, shot noise and fog. All the models can handle brightness and saturate well. To study the robustness of models against spatial and temporal corruptions, we present the mPC and rPC separately in Table 2. It shows that the models have similar rPC on spatial corruptions on both datasets, ranging from 67.9 ~ 76.5%.

**Robustness w.r.t temporal corruptions.** A trend can be observed that the robustness of models against temporal corruptions increases with the clean accuracy from Table 1. Table 2 also shows that the models with the best generalization have the highest mPC on temporal corruptions. Models trained on Mini Kinetics handle all types of temporal corruptions well. Interestingly, although Bit Error and Packet Loss can propagate and augment corruptions in the consequent frames, the performance of models for Mini Kinetics degrades less on these two corruptions, even if some frames are not recognizable by human. However, the models for Mini SSV2 are more sensitive to temporal corruptions. Especially, the accuracy drops by 21%, 13.2% and 10.4% on motion blur, bit error and frame rate conversion for Mini SSV2-C. It only drops by 14.1%,10.9% and 1.2% for Mini Kinetics-C, respectively. We also find that rPC of models on temporal corruptions for Mini SSV2 is 4~8% lower than them for Mini Kinetics in Table 2.

### 4.4 Q3: Trade off between Robustness, Generalization and Efficiency

As seen in Figure 2, when the clean accuracy of models improves, the mPC of models improves as well. On both datasets, we observe the same trend that rPC of models improves slightly when the clean accuracy improves. On Mini Kinetics-C, the clean accuracy, mPC and rPC of 3D ResNet50 are 10.4%,10.6% and 13.5% higher than those of X3D-M, respectively. The results show that models with better generalization are also more robust against the common corruptions. From the dimension of computation cost in Figure 3, the mPC increases with floating-point operations (FLOPs) per sample. On both datasets, the rPC decreases from 81% to 78% when the FLOPs of 3D ResNet-50 reduce from 180.2G to 46.5G FLOPs of SlowFast network with the backbone of ResNet50. TAM is an outlier in terms of rPC, as it is the only model using pure 2D CNN. Moreover, the claimed efficient approaches

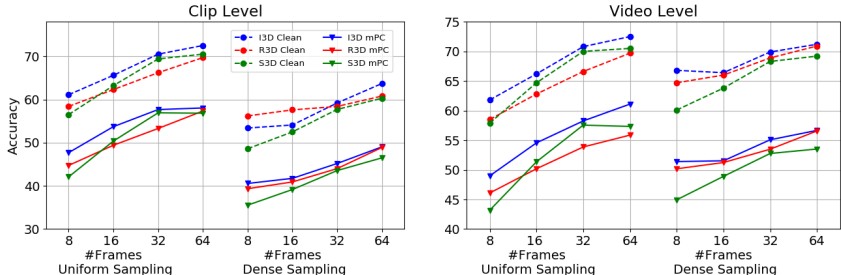

Figure 4: Robustness comparison of uniform sampling and dense sampling at clip level and video level on Mini Kinetics-C. For each setting, we use 8,16 ,32 , 64 frames as input. We use I3D, S3D and R3D(3D-ResNet) for training and evaluation. R3D use a backbone of ResNet18. The dash line indicates the accuracy of models on clean validation data. The solid line indicates the mPC of models on corrupted data.

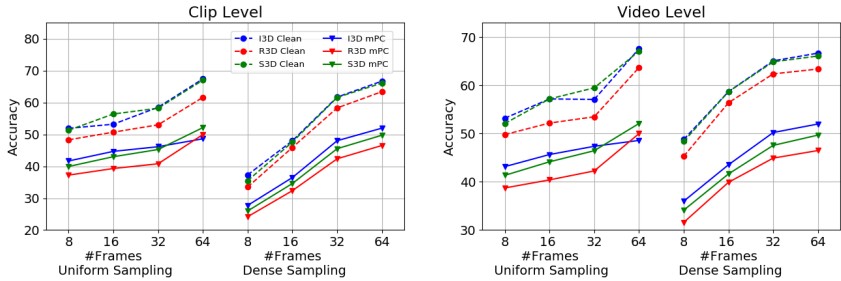

Figure 5: Robustness comparison of uniform sampling and dense sampling at different input frames on Mini SSV2-C. We use the same setting and networks as comparison on Mini Kinetics-C.

SlowFast, X3D-M and TAM have relatively lower rPC on temporal corruptions in Table 2. It appears that there is a trade-off between efficiency and corruption robustness.

### 4.5 Ablation Study

**2D-3D Module.** For comparing the corruption robustness of 2D and 3D models, we use InceptionV1 [38] and ResNet18 [15] as backbone, and input length of 8 and 16 at the clip level. With the same backbone, input length and training approach, the 3D models consistently outperform the 2D models in terms of rPC on both datasets, as shown in Table 3. On Mini Kinetics-C, the clean accuracy and mPC of 2D models remain the same when the input length increase, but the accuracy of 3D models increase by 4% ~ 6%. On Mini SSV2-C, the 3D models improve both clean accuracy and mPC by 18% ~ 22%. Table 1 shows that though TAM achieves the best clean accuracy, its rPC is 5~8% lower than the 3D models.

**Input Sampling Strategy.** As we introduced in the evaluation protocol, uniform sampling and dense sampling are widely used in video classification tasks. To assess the effect of sampling strategy on model corruption robustness, we train three classic video classification models I3D [2], 3D ResNet-18 [14] and S3D [45] using uniform sampling and dense sampling on both Mini Kinetics and Mini SSV2. In each setting, we use 8,16,32,64 frames as input. We also evaluate the generalization and robustness of models at clip and video levels. The clean accuracy and mPC of models on the benchmark are shown in Figure 4 and Figure 5.

Figure 4 shows that uniform sampling outperforms dense sampling on all settings at clip levels on Mini Kinetics-C. The mPC of models using uniform sampling are 4~13% higher than dense sampling. Because uniform sampling at clip level extracts the frame from the whole video evenly, it enables the model to capture longer-term temporal information than dense sampling at the clip level. At the video

Table 3: Clean Accuracy and mPC of 2D and 3D Models

| Frames | Backbone | Approach | Mini Kinetics-C | | | Mini SSV2-C | | |
|---|---|---|---|---|---|---|---|---|
| | | | Clean Acc | mPC | rPC | Clean Acc | mPC | rPC |
| 8 | InceptionV1 | 3D | 61.1 | 47.7 | **78.1** | 51.9 | 41.7 | **80.3** |
| | | 2D | 68.1 | 52.0 | 76.4 | 33.4 | 22.1 | 66.2 |
| 8 | ResNet18 | 3D | 58.4 | 44.9 | **76.9** | 48.3 | 37.2 | **77.0** |
| | | 2D | 66.9 | 50.5 | 75.5 | 30.3 | 18.6 | 61.4 |
| 16 | InceptionV1 | 3D | 65.6 | 53.7 | **81.9** | 53.2 | 44.7 | **84.0** |
| | | 2D | 67.8 | 52.7 | 77.7 | 34.6 | 22.7 | 65.6 |
| 16 | ResNet18 | 3D | 62.4 | 49.4 | **79.2** | 50.7 | 39.3 | **77.5** |
| | | 2D | 67.4 | 51.6 | 76.6 | 30.6 | 19.9 | 65.0 |

level, uniform sampling yields similar results to clip level. Dense sampling boosts by 9∼11% on mPC, comparing to clip level. However, all the models using uniform sampling perform better than dense sampling at frames of 32 and 64, except 3D ResNet-18 at 64 frames. On Mini SSV2-C, the uniform sampling of input makes model more robust than dense sampling at 8 frames and 16 frames at both clip and video levels. The mPC of models using different sampling methods are similar at 32 frames and 64 frames. Most of the videos from SSV2 dataset have 30 to 60 frames, both sampling methods will extract similar frames at input length larger than 32.

**Input Length.** From Figure 4 and Figure 5, we find the mPC of models increase with input frame length on both datasets. We also find the same trend of rPC from Table 3. The rPC of models using 16 frames is larger than models using 8 frames under all settings. It indicates that models using longer frames input with more temporal information can improve the robustness of models.

## 5  Robust Architecture, Training and Evaluation Protocols

We have conducted a large-scale evaluation on the robustness of CNN models towards common corruptions in video classification. Based on the study, we conclude several rules which are valid on both datasets in the benchmark. It provides guidance for deploying robust models from architecture, training and evaluation aspects. From the perspective of architecture, we systematically show that 3D CNN modules are more robust than 2D CNN modules. We also find that CNN models that improve inference efficiency may suffer from degradation of corruption robustness. From image classification and pose estimation tasks, it has been shown that networks with larger capacity can improve the robustness towards corruptions [16][42]. It is necessary to rethink the trend of model design when we introduce another dimension of robustness in video classification tasks. From training and evaluation aspects, uniform sampling outperforms dense sampling under most circumstances in terms of robustness. Moreover, uniform sampling enables the network to capture long-term temporal information with a less computational budget. Increasing input frame length can improve models generalization and robustness consistently, we may choose the input frame length as large as possible when we deploy the models in the wild. The rules of robust CNN models also imply that inputs and models which can capture more temporal information are more robust towards common corruptions. Similarly, we humans use temporal reasoning to tolerate the spatial corruptions generated in acquisition and processing [23].

## 6  Conclusion

In this paper, we have proposed a robust video classification benchmark (Mini Kinetics-C and Mini SSV2-C). It contains various types of temporal corruptions which distinguish from spatial level image corruptions in all image robustness benchmarks. With the proposed benchmarks, we have conducted an exhaustive large-scale evaluation on the corruption robustness of the state-of-the-art CNN-based models in video classification. Based on the evaluation results, we have drawn several conclusions in terms of robust model design, training and inference. We also provide some insights into the impact of temporal information on model robustness, which has been largely ignored in the existing research works. As a new dimension of research in video classification, robustness is to be systematically studied in future works due to its universality and significance.

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
