# OpenReview forum: "Benchmarking the Robustness of CNN-based Spatial-Temporal Models"
_NeurIPS.cc/2021/Track/Datasets_and_Benchmarks/Round1 — Submitted to NeurIPS 2021 Datasets and Benchmarks Track (Round 1)_

### Official Review · Reviewer_nset · 2021-06-26
**First benchmark and benchmarking study on corruption robustness of video classification models**

**Rating:** 5
**Confidence:** 4

**Strengths:**

* This is the first benchmark and benchmarking study on the corruption robustness of video classification models.

* The two datasets in the benchmark are reasonably large and rely differently on the spatial and temporal information, respectively, for video classification.


**Weaknesses:**

* This work is of more interest to the computer vision community than the general machine learning research community.

* It is not clear whether the 12 types of corruption and 5 levels of severity introduced to the datasets indeed faithfully reflect the common corruptions observed in real-world video classification applications. Moreover, the performance measure with which comparative study was performed is based on the assumption that the 60 ($12 \times 5$) corruption combinations are equally likely to occur. It is likely that the findings of the comparative study, and hence the guidance for robust model design, can be dramatically different if the corruption distribution is changed.

* Since each of the 12 types of corruption is either a spatial corruption or a temporal corruption, a corrupted video can only involve corruption of either one of the two categories, but not both. However, in reality, it is possible for a corrupted video to involve both spatial and temporal corruptions.

* This study only trains the models on clean videos and evaluates them on corrupted videos. It is well known in the machine learning community that training models on noisy (corrupted) data can usually help to improve the generalization capability and robustness of the models. This aspect is however ignored in the current study.


**Additional Feedback:**

Post-rebuttal comments:
I thank the authors for responding to my comments, but I still think that this work is not of broad interest to the NeurIPS community.

**Clarity:**

There are language errors throughout the paper. There is room for improvement.


**Correctness:**

The mean performance under corruption (mPC), also the rPC which is defined based on mPC, assumes that all corruption types and all severity levels are equally likely to occur. Is this assumption correct? More generally, how can you ensure that the corruption distribution $P_C$ correctly reflects the common corruptions in the real world or the nature?


**Documentation:**

Description and code are provided for generating the corrupted videos.


**Ethics:**

I am not aware of any potential ethical or social implication.


**Relation To Prior Work:**

This work can be seen as generalizing the study of corruption robustness in image classification to video classification. In particular, the study of spatial corruptions in videos is closely related to that of corruptions in images. Related papers for image classification are cited.


**Summary And Contributions:**

This work seeks to establish a benchmark for assessing the robustness of existing 2D and 3D CNN-based spatial-temporal models for video classification. Although there has been much development in improving the robustness of models for image classification, that for video classification involving temporal corruptions in addition to spatial corruptions is much less explored. Based on the benchmark which consists of two corrupted versions of (subsets of) two existing large-scale video datasets, the benchmarking study presented in the paper sheds light on the design, training and inference of robust models for video classification. It is the first attempt of its kind and may lead to follow-up work by others in the video understanding research community.

---

### Official Review · Reviewer_SQA7 · 2021-07-03
**Potentially useful dataset but needs more extensive evaluation and comparison against previous literature**

**Rating:** 6
**Confidence:** 3

**Strengths:**

1. Temporal corruptions for video classification: while there are works that deal with introducing spatial corruptions to images/videos (frames), a missing but important direction in testing model robustness is the introduction of temporal noise. In video understanding models, this helps in evaluating how well the models are leveraging temporal information for the assigned task. This paper attempts a step in that direction by introducing certain forms of temporal corruption in 2 standard video datasets, which have some complementary attributes in that they require different levels of video understanding for succesful activity recognition. Such work might be able to motivate more future research that focuses on improved and more extensive datasets for adding temporal distortions and consequently, more robust and stronger models that better leverage temporal information in videos.

2. Extensive evaluation and comparison of standard video classification models: the paper also comprehensively compares different well-known video classification architectures, frame sampling and learning strategies. Such an evaluation helps uncover certain helpful but sometimes obvious insights about architecture design, sampling strategy and training methodologies that could help in future research in the design of better video classification models.

**Weaknesses:**

1. Lack of related work on video noise: since there is already a very matured body of work on video processing, I believe there should be some literature that describes the most common forms of noise that occurs in videos. Such a link would allow for a better understanding of the temporal distortions in particular and the distribution of their occurence in the real world. That would make the claims in the paper regarding the corrupted datasets following the real-world distribution of noises in the dataset well-substantiated and the paper stronger.

2. Lack of experiment that shows training with noise: a pretty important evaluation setting, in my opinion, is where the models are trained and evaluated with noise. If the real-world noise distribution is well-known, I don't see why one shouldn't allow the models to have access to noisy data during trainining. Such training might provide additional insights on the behaviors and learned representation of differenent models when they are trained with noise.

3. Missing comparison with state-of-the-art transformer-based models: AFAIK, most current state-of-the-art models are transformer-based. So it's pretty important to evaluate such models on these datasets to get an idea of how effective the different corruptions and these datasets, as a whole, are.

P.S.: I am willing to increase my rating if the above-listed concerns addressed.

**Additional Feedback:**

1. Page 7, 4.3 Q2 'Robustness wrt spatial corruptions': 'saturate' should be 'saturation'

2. rPC: I am not sure how much value this metric adds to the evaluation. I feel that absolute mPC conveys the full picture pretty well. An example of a scenario where rPC is misleading: for a very strong model, rPc might actually look bad just because the normalizing factor value for such a model might be very hight but in absolute terms, that model might be more robust than any other model.

**Clarity:**

The paper is moderately very clearly.
Some issues:
1. Page 4, text above equation 1: both corruption type and corrupted data distributions use the same notation. Maybe change one for better clairty.

2. Page 5, description of Mini Kinetics-C: not sure what 'infusion' means in this context

**Correctness:**

The claims made in the paper and the supporting experiments/evaluation methods look somewhat correct to me. Possible issues are listed as part of weakness.

**Documentation:**

The dataset looks fairly well-documented but it would help a user if the readme of the github repositiory is improved by describing what the main pieces of code are and how to use them.

**Ethics:**

I couldn't spot any ethical issues.

**Relation To Prior Work:**

The work compares with prior work somewhat extensivelyand tries to address their shortcomings. However, some more comparisons and evaluation would make the work even stronger. I have listed such possible comparisons as part of 'Weakness'.

**Summary And Contributions:**

This work presents the corrupted versions of two standard video classification datasets with somewhat complementary characteristics, where the corruptions try to capture the real-world behavior in terms of types of corruption and their distribution. The corruptions introduced are both at the spatial and temporal level of the videos, thus adding another important dimension in the testing of the robustness of video understanding models. The authors systematically benchmark standard video classification models on these datasets and unearth some insights regarding architectural properties that look important vis-a-vis motivating future research in building video recognition architectures.

Edit: rating updated on July 13th

---

### Official Review · Reviewer_wuy3 · 2021-07-04
**Interesting study but missing some key evaluations**

**Rating:** 7
**Confidence:** 5
**Correctness:** Correct.
**Clarity:** Yes, this paper is well written.

**Strengths:**

1. This paper is the first to systematically study the robustness of video classification models towards corruptions.
2. Extensive experiments have been conducted to study various factors that affect model robustness.
3. Inspiring conclusions have been made based on the experiments.
4. This paper is well written.

**Weaknesses:**

1. Missing study about robustness with respect to levels of corruptions.  As presented in the paper, there are 5 levels for each corruption type. All the reported results are averaged over all the levels. In this way, it is unclear how the performance varies in different corruption levels.

2. Missing more ablation study. Different models may vary the performance with different hyper-parameters, e.g., batch size, training epochs, etc. It may not be a good practice to use the same hyper-parameters for all methods, as the settings may be biased towards some models.

3. How about training the models with corrupted data. The models are trained with clean data and tested on corrupted data. How about training the model on corrupted data to encourage generalizability by adding some regularizations, e.g., consistency regularization, like many recent self-supervised learning methods did.

4. Figure 1 is not referred in the text.

**Additional Feedback:**

Post-rebuttal comments: The rebuttal has successfully addressed my concerns. So, I would like to raise my score.

**Documentation:**

Yes.

**Ethics:**

No.

**Relation To Prior Work:**

Yes, clear relationship to prior work is discussed.

**Summary And Contributions:**

This paper presents a systematic study on the robustness of video classification models towards spatial and temporal corruptions. Two benchmarking datasets are proposed, which are constructed by adding 60 types of spatial and temporal corruptions on the clean images, and several well-established video classification models are evaluated on the corrupted benchmarks. Based on the study, several conclusions are drawn, which could inspire future robust model design towards corruptions for the task.

---

### Decision · Program_Chairs · 2021-07-26

**Decision:**

Reject

**Comment:**

The paper poses a difficult decision since the reviewers have diverging views on whether the paper should be accepted. One concern raised by multiple reviewers is whether the corruptions studied in the paper are realistic representations of the types of noise encountered in video data. Here, an interesting contribution is that the authors experiment with various noise types arising from video compression. On the other hand, it is less clear how the video acquisition corruptions such as the synthetic weather patterns (fog, rain, etc.) relate to corruptions encountered in the real world. For static images, this has been the subject of prior work (https://arxiv.org/abs/2007.00644) who found that natural distribution shifts behave differently from synthetic image corruptions. While this does not invalidate the use of synthetic image corruptions, it raises the question whether the situation is different for noise arising in videos.

Moreover, the submission claims to "make the first attempt to conduct an exhaustive study on corruption robustness in terms of spatial and temporal domain", but does not discuss its relationship to relevant earlier work that leveraged video data to study robustness, in particular https://arxiv.org/abs/1904.10076 and https://arxiv.org/abs/1906.02168 . These paper specifically use video data as a source of perturbations arising naturally.

Overall I find the direction of the proposed benchmark interesting: neural networks are not as robust as we would like them to be, and robustness to perturbations in videos has been studied less than robustness to perturbations in static images. But the current submission does not take relevant prior work into account, and could be improved by clarifying how the findings from the proposed benchmark relate to existing robustness benchmarks in the single-image domain. Hence I recommend rejecting the paper from the first round of the datasets & benchmarks track, but encourage the authors to take the reviewer comments into account and submit an updated version of their paper to the second round of the datasets & benchmarks track or to another venue.